# Efficient Bayesian Updates for Deep Active Learning via Laplace Approximations

## Abstract

Deep active learning (AL) involves selecting batches of instances for annotation since retraining large deep neural networks (DNNs) after each label acquisition is computationally impractical. Employing a naive top-$b$ selection can result in a batch of redundant (similar) instances. To address this issue, various batch AL strategies have been developed, many of which employ clustering for diversity as a heuristic. In contrast, we approach this issue by substituting the costly retraining with an efficient Bayesian update. Our proposed update represents a second-order optimization step using the Gaussian posterior from a last-layer Laplace approximation. Thereby, we achieve low computational complexity by computing the inverse Hessian in closed form. We demonstrate that in typical AL settings, our update closely approximates retraining while being considerably faster. Leveraging our update, we introduce a new framework for batch selection through sequential construction by updating the DNN after each label acquisition. Furthermore, we incorporate our update into a look-ahead selection strategy as a feasible upper baseline approximating optimal batch selection. Our results highlight the potential of efficient updates to advance deep AL research.

## 1 Introduction

Active Learning (AL) sequentially selects instances for annotation by human experts, aiming to maximize model performance while minimizing labeling efforts. When combined with deep neural networks (DNNs), AL typically selects instances in batches rather than one at a time. The reason for this is that retraining DNNs after each label acquisition is computationally expensive, and delay can lead to additional costs since annotators' time is valuable (Kirsch et al., 2023).

In a naive top-$b$ batch selection, a batch of $b$ instances with the highest scores is chosen based on an informativeness measure. However, when many similar instances are present, this approach can result in significant redundancy within the batch (similar instances tend to receive similarly high scores). To address this issue, many selection strategies have been developed to replace this naive selection (Ren et al., 2021). These strategies often employ clustering techniques to select diverse batches, ensuring that instances within a batch are dissimilar to one another (Hacohen et al., 2022). While effective in reducing redundancy, clustering does not guarantee optimal selection due to its heuristic motivation.

Orthogonal to these strategies, we explore the concept of efficient "retraining" in deep AL. If retraining were computationally feasible, researchers could place greater emphasis on the development of theoretically sound informativeness measures instead of using heuristic clustering approaches to ensure diversity. Additionally, selection strategies that aim to maximize future performance–strategies that have been shown to be near-optimal in traditional AL (Roy & McCallum, 2001)–could be made feasible with DNNs. Therefore, we examine the concept of updating DNNs through a single optimization step as a proxy for retraining and explore its potential to enhance the AL process.

To underscore the requirements of such an update, consider strategies designed to maximize future performance. Typically, these use a look-ahead to select instances that significantly change model predictions. More specifically, they examine how adding unlabeled instances to the labeled pool and retraining the model affects predictions (Roy & McCallum, 2001). However, with a large number of unlabeled instances and the time-consuming retraining process of DNNs, this approach becomes infeasible. Therefore, instead of retraining, a highly efficient update method is required. The only work to realize such an approach with DNNs is by Tan et al. (2021), which employ an ensemble of

DNNs combined with Monte Carlo (MC) updates via Bayes' theorem. Although this update makes a one-step look-ahead feasible, it remains suboptimal for several reasons: (i) the update requires an ensemble of DNNs, making the actual retraining time and memory demands inefficient; (ii) the update does not accurately reflect the performance of full retraining; and (iii) the updating process becomes inefficient with an increasing number of ensemble members.

In this article, we propose an efficient update method for DNNs in the context of AL for classification. Specifically, we transform an arbitrary DNN into a Bayesian neural network (BNN) by employing a last-layer Laplace approximation (LA) (Daxberger et al., 2021). While the closed-form expression of the posterior allows us to leverage second-order optimization techniques, we ensure low computational complexity by computing the required inverse Hessian analytically. Unlike the MC-based update used in (Tan et al., 2021), our approach does not require an ensemble of DNNs, making it easily applicable and both memory- and training-efficient (Daxberger et al., 2021). Additionally, because we utilize a single DNN, we can leverage pretrained foundation models (Oquab et al., 2023), which have been shown to be an essential part of many deep AL strategies (Hacohen et al., 2022; Gupte et al., 2024). The resulting update is fast and closely matches the performance of full retraining. Extensive studies across different data modalities, including image and text datasets, demonstrate that our updates outperform the typically employed MC-based ones (Tan et al., 2021) in terms of speed and performance. Furthermore, we examine the proposed update in two distinct AL scenarios:

1. **Enhancing Existing Strategies with Immediate Label Utilization:** We propose a simple framework to improve existing strategies by immediately making use of acquired labels through the proposed updates. Rather than selecting the top-$b$ highest-scoring instances simultaneously, we iteratively select the highest-scoring instance $b$ times *but* update the model between each selection. This simple strategy, which approximates single-instance AL during batch construction, performs surprisingly well, outperforming naive top-$b$ selection as well as selection strategies that employ clustering.

2. **Optimal AL with Look-Ahead Selection:** We investigate the potential of our update with a look-ahead selection strategy in an optimal AL setting. Specifically, we approximate an optimal selection strategy that maximizes future performance. Instead of retraining, we ensure computational feasibility by employing our update. The resulting strategy outperforms all competitors, showcasing that currently employed selection strategies have much potential for improvement.

---

**Summary of Contributions**

- **Efficient DNN Update:** We propose an efficient update method for DNNs that employs a Laplace approximation and second-order optimization techniques. We enable low computational complexity through closed-form computation of the inverse Hessian.

- **Comprehensive Evaluation:** We perform an extensive evaluation across data modalities, demonstrating that our update outperforms MC-based updates in both speed and accuracy.

- **Immediate Label Utilization:** We develop a simple framework that employs our update to immediately incorporate acquired labels, improving existing selection strategies by updating the model during batch construction.

- **Optimal AL with Look-Ahead:** We study our update in an optimal AL setting, making a near-optimal selection strategy as an upper baseline computationally feasible.

---

## 2 RELATED WORK

**Pool-based deep AL** selection strategies are typically divided into uncertainty-based, diversity-based, and hybrid strategies. *Uncertainty-based strategies* assume difficult-to-classify instances as beneficial. Margin sampling (Settles, 2009), a popular variant (Bahri et al., 2022), selects instances where the difference between the two highest predicted class probabilities is largest. Bald (Gal et al., 2017) assumes a BNN and selects instances that maximize the mutual information between class predictions and the DNN's posterior distribution. Due to the requirement of batch acquisition in deep AL, these strategies typically select the top-$b$ highest-scoring instances. *Diversity-based strategies* assume that a set of diverse instances benefits the model. Core-Set (Sener & Savarese, 2018) selects instances that

minimize the average distance between the feature representations of labeled and unlabeled instances. In practice, *hybrid strategies*, a combination of both uncertainty and diversity, have been shown to work well. BatchBALD (Kirsch et al., 2019) extends BALD by reducing redundant information within a batch. Badge (Ash et al., 2020) selects instances with high gradient norms based on pseudo-labels and ensures diversity by employing $k$-MEANS++ in the gradient space. Typiclust (Hacohen et al., 2022) replaces the notion of uncertainty with typicality and selects typical instances from clusters obtained through $k$-MEANS.

**Look-ahead strategies** (Roy & McCallum, 2001; Kottke et al., 2021) remain relatively underexplored in deep AL. These strategies aim to select instances expected to improve the model's performance the most by retraining for all possible candidate instances. In non-deep settings, such approaches have been shown to achieve near-optimal selection (Roy & McCallum, 2001) while offering convergence guarantees (Zhao et al., 2021). However, adapting these strategies to deep AL is challenging due to the computational cost of retraining. To the best of our knowledge, BEMPS (Tan et al., 2021) is the only strategy to implement a look-ahead mechanism in deep AL. They employ deep ensembles and MC-based updates via Bayes' theorem (see Section 3). While this update is computationally efficient, its performance falls short compared to full model retraining.

Similar to our setting, **continual learning** (De Lange et al., 2021) updates models by exclusively training with data from a new task, addressing the challenge of retaining knowledge from previously learned tasks. Popular techniques (Kirkpatrick et al., 2017; Ritter et al., 2018a) use conventional first-order optimization methods, incorporating a regularization term to counteract catastrophic forgetting. Specifically, Ritter et al. (2018a) and Kirkpatrick et al. (2017) derive a regularization term from an LA that penalizes large deviations from prior knowledge. Unlike our method, these approaches require training over multiple epochs for the regularization term to have an impact. Once it is used as an update (i.e., single optimization step), these strategies simplify to a first-order gradient step. Additionally, they assume large amounts of new data per task (thousands of instances), whereas our update method is designed for small datasets, ranging from a single to hundreds of instances. For example, a typical benchmark is to extend a dataset with a task consisting of all instance-label pairs of a new class (ca. 5,000 in MNIST).

More closely related to our work is **online learning** (Hoi et al., 2021), which aims to sequentially and efficiently update models from incoming data streams. Traditional approaches often focus on linear (Zinkevich, 2003; Crammer et al., 2006) or shallow (Kivinen et al., 2001; Sahoo et al., 2014) models with maximum-margin classification. However, applying online learning to DNNs remains difficult due to issues such as convergence, vanishing gradients, and large model sizes (Sahoo et al., 2018; Yoon et al., 2018). To address these challenges, Sahoo et al. (2018) proposed a method that modifies a DNN's architecture to facilitate updates. We argue that this approach is restrictive in state-of-the-art settings, given the increasing reliance on pretrained foundation models (Devlin et al., 2019; Oquab et al., 2023). Most similar to our setting is the work on Bayesian online inference by Kirsch et al. (2022), which is also employed in (Tan et al., 2021). The core idea is to sample hypotheses, e.g., via MC-Dropout, from the posterior distribution of a BNN and weight their importance according to the respective likelihoods for sequentially arriving data. The empirical results raised concerns regarding the applicability of such updates in high-dimensional parameter spaces. We refer to these updates as MC-based updates.

**BNNs** (Wang & Yeung, 2020; Fortuin, 2022) induce a prior distribution on the parameters of a DNN and learn a posterior distribution given data. MC-Dropout (Gal & Ghahramani, 2016) uses dropout during inference to obtain a distribution over predictions. While it is simple to use, its inference is inefficient, and it provides suboptimal uncertainty estimates (Ovadia et al., 2019). Deep ensembles (Lakshminarayanan et al., 2017) are known for their superior uncertainty estimates but are train and memory inefficient (Ovadia et al., 2019). LAs (Ritter et al., 2018b) approximate the posterior as a Gaussian, with the MAP estimate as the mean and the inverse Hessian as the covariance. As computing this Hessian is expensive for large DNNs, LA is often used only in the last layer (Daxberger et al., 2021).

## 3 FAST BAYESIAN UPDATES FOR DEEP NEURAL NETWORKS

In this section, we present our new update method. First, we introduce the general concept of Bayesian updates together with the variant MC-based updates (Kirsch et al., 2022; Tan et al., 2021).

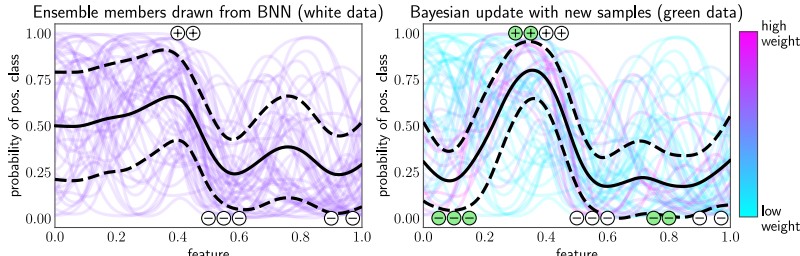

Figure 1: The left plot shows the predicted probabilities of the positive class for each hypothesis (colored lines) drawn from a BNN as well as the mean (black solid line) and standard deviation (black dashed line) of its predictive distribution. The right plot shows updated weights for each hypothesis and the predictive distribution after observing additional instances (green).

Afterward, we propose our novel method focusing on an efficient update of the Gaussian posterior distribution via last-layer LAs. For an introduction to LA, we refer to (Daxberger et al., 2021).

### 3.1 BAYESIAN UPDATES

We focus on classification problems with instance space $\mathcal{X}$ and label space $\mathcal{Y} = \{0, \ldots, K-1\}$. The primary goal in our setting is to efficiently incorporate the information of new instance-label pairs $\mathcal{D}^{\oplus} = \{(\boldsymbol{x}_n, y_n)\}_{n=1}^N \subset \mathcal{X} \times \mathcal{Y}$ into a DNN trained on dataset $\mathcal{D} \subset \mathcal{X} \times \mathcal{Y}$. Retraining the entire network on the extended dataset $\mathcal{D} \cup \mathcal{D}^{\oplus}$ results in high computational cost for a large dataset $\mathcal{D}$. Conversely, using the new data solely can cause catastrophic forgetting (Ritter et al., 2018a).

For this purpose, we employ BNNs with Bayesian updates (Opper & Winther, 1999) as an efficient alternative to retraining. The main idea of BNNs is to estimate the posterior distribution $p(\boldsymbol{\omega}|\mathcal{D})$ over the parameters $\boldsymbol{\omega} \in \Omega$ given the observed training data $\mathcal{D}$ using Bayes' theorem. The obtained posterior distribution over the parameters can then be used to specify the predictive distribution over a new instance's class membership via marginalization:

$$p(y|\boldsymbol{x}, \mathcal{D}) = \mathbb{E}_{p(\boldsymbol{\omega}|\mathcal{D})}[p(y|\boldsymbol{x}, \boldsymbol{\omega})] = \int p(y|\boldsymbol{x}, \boldsymbol{\omega})p(\boldsymbol{\omega}|\mathcal{D}) \, \mathrm{d}\boldsymbol{\omega}. \tag{1}$$

Thereby, the likelihood $p(y|\boldsymbol{x}, \boldsymbol{\omega}) = [\mathrm{softmax}(f^{\boldsymbol{\omega}}(\boldsymbol{x}))]_y$ denotes the probabilistic output of a DNN with parameters $\boldsymbol{\omega}$, where $f^{\boldsymbol{\omega}} : \mathcal{X} \to \mathbb{R}^K$ is a function outputting class-wise logits.[1]

The formulation in equation 1 provides a theoretically sound way to obtain updated predictions. In particular, this is because the probabilistic outputs $p(y|\boldsymbol{x}, \boldsymbol{\omega})$ do not directly depend on the training data $\mathcal{D}$. Consequently, to obtain an updated predictive distribution, we do not need to update the parameters $\boldsymbol{\omega}$ directly but only the posterior distribution $p(\boldsymbol{\omega}|\mathcal{D})$. The updated posterior distribution $p(\boldsymbol{\omega}|\mathcal{D}, \mathcal{D}^{\oplus})$ is found through Bayes' theorem, where the current posterior distribution $p(\boldsymbol{\omega}|\mathcal{D})$ is considered the prior and multiplied with the likelihood $p(y|\boldsymbol{x}, \boldsymbol{\omega})$ per instance-label pair $(\boldsymbol{x}, y) \in \mathcal{D}^{\oplus}$. As instances in $\mathcal{D}$ and $\mathcal{D}^{\oplus}$ are assumed to be independently distributed, we can simplify the likelihood and reformulate the parameter distribution as follows[2]:

$$p(\boldsymbol{\omega}|\mathcal{D}^{\oplus}, \mathcal{D}) \propto p(\boldsymbol{\omega}|\mathcal{D})p(\mathcal{D}^{\oplus}|\mathcal{D}, \boldsymbol{\omega}) \overset{\text{i.i.d.}}{=} p(\boldsymbol{\omega}|\mathcal{D})p(\mathcal{D}^{\oplus}|\boldsymbol{\omega}) = p(\boldsymbol{\omega}|\mathcal{D}) \prod_{(\boldsymbol{x}, y) \in \mathcal{D}^{\oplus}} p(y|\boldsymbol{x}, \boldsymbol{\omega}). \tag{2}$$

We refer to equation 2 as the *Bayesian update*.

The most common realization (Kirsch et al., 2022; Tan et al., 2021) of this update is through MC-based BNNs, such as MC-Dropout and deep ensembles. These BNNs rely on samples (or hypotheses) $\boldsymbol{\omega}_1, \ldots, \boldsymbol{\omega}_M$ drawn from an approximate posterior $q(\boldsymbol{\omega}|\mathcal{D})$. Research (Yoon et al., 2013; Tan et al., 2021) assumes that all hypotheses are equally likely to explain the observed data and have the same probability before updating. By updating the posterior distribution through equation 2, they weigh more likely hypotheses given the new data higher. We refer to these as MC-based updates with a

---

[1]We denote the $i$-th element of a vector $\boldsymbol{b}$ as $[\boldsymbol{b}]_i = b_i$.
[2]We denote $p(y_1, \ldots, y_N \mid \boldsymbol{x}_1, \ldots, \boldsymbol{x}_N, \boldsymbol{\omega})$ with $\mathcal{D} = \{(\boldsymbol{x}_n, y_n)\}_{n=1}^N$ as $p(\mathcal{D}|\boldsymbol{\omega})$.

formal definition given in Appendix A. Figure 1 illustrates this concept where different hypotheses $\boldsymbol{\omega}_1, \ldots, \boldsymbol{\omega}_M \sim q(\boldsymbol{\omega}|\mathcal{D})$ are shown. Each hypothesis represents a possible true solution for the learning task (white instances). When new data (green instances) arrives, we weigh each hypothesis by its likelihood of explaining the new data and obtain an updated prediction without retraining. This results in an updated predictive distribution, as seen in bold in Figure 1 (right).

## 3.2 FAST APPROXIMATIONS OF BAYESIAN UPDATES FOR DEEP NEURAL NETWORKS

Our update method is based on a combination of two concepts. First, instead of MC-based BNNs, we suggest using LAs on the last layer of a DNN. Second, we directly modify the approximate posterior distribution of the LA, providing a much more flexible way to adapt it to new data than reweighting. In the following, we explain each component in detail. For now, we focus on binary classification with $K = 2$, and refer to Appendix C for an extension to multi-class classification.

**Last-layer LA**: LAs approximate the (intractable) posterior distribution $p(\boldsymbol{\omega}|\mathcal{D})$ with a Gaussian centered on the maximum a posteriori (MAP) estimate with a covariance equal to the negative Hessian of the log posterior (Daxberger et al., 2021). We denote this approximate distribution as

$$q(\boldsymbol{\omega}|\mathcal{D}) = \mathcal{N}(\boldsymbol{\omega}|\hat{\boldsymbol{\mu}}, \hat{\boldsymbol{\Sigma}}) \propto q(\boldsymbol{\omega}) \prod_{(\boldsymbol{x}, y) \in \mathcal{D}} p(y|\boldsymbol{x}, \boldsymbol{\omega}), \tag{3}$$

where $q(\boldsymbol{\omega})$ is a Gaussian prior distribution. The MAP estimate $\hat{\boldsymbol{\mu}}$ results from training on $\mathcal{D}$ with conventional gradient optimization techniques. The covariance matrix $\hat{\boldsymbol{\Sigma}}$ is the inverse Hessian of the negative log posterior evaluated at the MAP estimate $\hat{\boldsymbol{\mu}}$ given training data $\mathcal{D}$. We model the posterior distribution only on the last layer of a DNN to ensure fast inference.

The benefits of using a last-layer LA are manifold. Given access to $q(\boldsymbol{\omega}|\mathcal{D})$ through a Gaussian, we enable *more flexible updates* compared to MC-based ones, as we can directly modify the mean and covariance. In contrast, MC-based updates only change the approximate distribution by reweighting hypotheses, leading to a strong dependency on the samples $\boldsymbol{\omega}_1, \ldots, \boldsymbol{\omega}_M$. Last-layer LAs can be *integrated seamlessly* into nearly all DNNs, including pretrained models, as only the covariance has to be computed to obtain $q(\boldsymbol{\omega}|\mathcal{D})$. This is particularly important in deep AL, where recent findings highlight self-supervised learning as a crucial factor in selecting informative instances (Hacohen et al., 2022; Gupte et al., 2024). Finally, compared to deep ensembles and MC-Dropout, last-layer LAs introduce *minimal computational overhead*. While deep ensembles require longer training and MC-dropout impairs the inference time, LAs simply need to calculate a covariance matrix after training and allow fast inference (cf. equation 1) through techniques such as mean-field approximation (Lu et al., 2020).

**Second-Order Update:** The second concept focuses on the update step of the Gaussian distribution. Observing new data, we follow the same approach as in equation 3, but with $q(\boldsymbol{\omega}|\mathcal{D})$ as our prior:

$$q(\boldsymbol{\omega}|\mathcal{D}, \mathcal{D}^{\oplus}) = \mathcal{N}(\boldsymbol{\omega}|\hat{\boldsymbol{\mu}}^{\text{upd}}, \hat{\boldsymbol{\Sigma}}^{\text{upd}}) \propto q(\boldsymbol{\omega}|\mathcal{D}) \prod_{(\boldsymbol{x}, y) \in \mathcal{D}^{\oplus}} p(y|\boldsymbol{x}, \boldsymbol{\omega}), \tag{4}$$

where $\hat{\boldsymbol{\mu}}^{\text{upd}}$ and $\hat{\boldsymbol{\Sigma}}^{\text{upd}}$ represent the updated mean and covariance, respectively. The resulting updated posterior $q(\boldsymbol{\omega}|\mathcal{D}, \mathcal{D}^{\oplus})$ is non-Gaussian due to $p(y|\boldsymbol{x}, \boldsymbol{\omega})$ being a categorical likelihood. Consequently, the closed-form computation of the integral in equation 1 becomes intractable. The basic idea of our update is to approximate the new posterior $q(\boldsymbol{\omega}|\mathcal{D}, \mathcal{D}^{\oplus})$ by first applying a second-order optimization step via Gauss-Newton and then estimating the new covariance at that point. Thus, the updated mean and covariance are given by:

$$\hat{\boldsymbol{\mu}}^{\text{upd}} = \hat{\boldsymbol{\mu}} - \gamma \boldsymbol{H}^{-1}(\hat{\boldsymbol{\mu}}, \hat{\boldsymbol{\Sigma}}, \mathcal{D}^{\oplus}) \sum_{(\boldsymbol{x}, y) \in \mathcal{D}^{\oplus}} (p_{\boldsymbol{x}} - y) \boldsymbol{h}_{\boldsymbol{x}}, \qquad \hat{\boldsymbol{\Sigma}}^{\text{upd}} = \boldsymbol{H}^{-1}(\hat{\boldsymbol{\mu}}^{\text{upd}}, \hat{\boldsymbol{\Sigma}}, \mathcal{D}^{\oplus}), \tag{5}$$

where $\boldsymbol{h}_{\boldsymbol{x}}$ denotes the representation of $\boldsymbol{x}$ at the penultimate layer, $p_{\boldsymbol{x}} = \text{sigmoid}(\boldsymbol{h}_{\boldsymbol{x}}^{\text{T}} \boldsymbol{\mu})$ is the probability for the positive class, and $\gamma$ is a factor controlling the step size. The required updated Hessian can be computed efficiently in closed form following Spiegelhalter & Lauritzen (1990) by

$$\boldsymbol{H}^{-1}(\boldsymbol{\mu}, \boldsymbol{\Sigma}, \mathcal{A}) = \boldsymbol{\Sigma} - \sum_{(\boldsymbol{x}, y) \in \mathcal{A}} \frac{p_{\boldsymbol{x}}(1 - p_{\boldsymbol{x}})}{1 + \overline{\sigma}_{\boldsymbol{x}} \cdot p_{\boldsymbol{x}}(1 - p_{\boldsymbol{x}})} \big(\boldsymbol{\Sigma} \boldsymbol{h}_{\boldsymbol{x}}\big)\big(\boldsymbol{\Sigma} \boldsymbol{h}_{\boldsymbol{x}}\big)^{\text{T}}, \tag{6}$$

where $\overline{\sigma}_{\boldsymbol{x}} = \boldsymbol{h}_{\boldsymbol{x}}^{\mathrm{T}} \boldsymbol{\Sigma} \boldsymbol{h}_{\boldsymbol{x}}$ is the predictive variance. The derivation can be found in Appendix D.

The idea behind using second-order optimization techniques is that they are more robust than first-order gradient optimization techniques due to the incorporation of curvature information of the log posterior. This results in a more accurate representation of the loss landscape, enabling more efficient and robust parameter updates that are less sensitive to hyperparameter choices. A critical aspect of our method's efficiency is that we do not need to recompute the Hessian from scratch. Instead, our updates leverage the covariance available through LAs and use the Woodbury identity (Woodbury, 1950) for closed-form inversion, significantly reducing computational overhead. Further, a common problem with last-layer LAs is that the Hessian can become a bottleneck when dealing with a large number of classes. To address this, we can approximate the Hessian in equation 6 by considering a Gaussian likelihood instead of a multi-class one, as also done in (Liu et al., 2023; Fortuin, 2022). Lastly, we want to highlight that an assumption of an LA is that we are at the mode of a distribution, and adding more data violates this assumption. As we focus on AL and only update with a few (up to hundreds) instances at a time, this issue is less severe. Empirically, this is also confirmed by our experiments in the next section.

## 4 BAYESIAN UPDATING EXPERIMENTS

In this section, we evaluate the efficiency of the proposed update by comparing it against competitors on various benchmark datasets for image and text classification. Our code is publicly available at https://github.com/anonymous/authors.

### 4.1 EXPERIMENTAL SETUP

Our **experimental design** is based on the work of Kirsch et al. (2022). First, we train a DNN on the training dataset $\mathcal{D}$ (baseline). We then use this baseline DNN to evaluate a last-layer LA and related Bayesian updates on additional instance-label pairs $\mathcal{D}^{\oplus}$ and compare these results to retraining the DNN on the complete dataset $\mathcal{D} \cup \mathcal{D}^{\oplus}$. We evaluate (i) the influence of the step size $\gamma$ on chosen validation datasets, (ii) the impact of our update at different learning stages of the DNN, (iii) the impact of our update with increasing sizes of new arriving datasets, and (iv) the time efficiency of our update by considering the speed-up factor against retraining. For comparison, we consider MC-based updates by sampling 10k hypotheses from the approximate Gaussian posterior $q(\boldsymbol{\omega}|\mathcal{D})$ and the less complex first-order updates only considering gradients. Note that the latter is equivalent to the continual learning strategy of (Ritter et al., 2018a), as we demonstrate in Appendix A. Since first-order updates do not use the Hessian, this comparison also allows us to assess the benefits of using second-order optimization. We exclude retraining solely on $\mathcal{D}^{\oplus}$, as we empirically found that it leads to catastrophic forgetting (Kirkpatrick et al., 2017). All performance metrics are averaged across 10 repetitions. For visual clarity, we do not report standard errors.

The datasets $\mathcal{D}$ and $\mathcal{D}^{\oplus}$ are randomly sampled from real-world datasets. We use three image and three text **benchmark datasets** commonly used in literature (Hacohen et al., 2022; Rauch et al., 2023) with varying complexity reflected through different numbers of classes. Table 1 gives an overview. A detailed summary for each dataset is provided in Appendix E.

Table 1: Overview of datasets.

| Type | Dataset | Reference | # classes |
|---|---|---|---|
| Image | Cifar-10 | (Krizhevsky, 2009) | 10 |
| | Snacks | (Matthijs, 2021) | 20 |
| | DTD | (Cimpoi et al., 2014) | 49 |
| Text | DBPedia | (Auer et al., 2007) | 14 |
| | Banking-77 | (Casanueva et al., 2020) | 77 |
| | Clinc-150 | (Larson et al., 2019) | 150 |

The goal of an update method is to ensure both effectiveness and speed. To assess this, we use different **performance metrics**. To evaluate *effectiveness*, or how well an update or retraining generalizes, we measure accuracy. When experimenting with hyperparameters, accuracy is assessed on a 10% validation split. Otherwise, it is measured on the test dataset. An optimal update method should achieve the same performance as completely retraining the DNN with $\mathcal{D} \cup \mathcal{D}^{\oplus}$. To assess the *speed* of an update, we report the speed-up factor compared to retraining by dividing the time required for retraining by the time required for updating (equation 5 and 6). Retraining and updating times were recorded on an NVIDIA RTX 4090 GPU and an AMD Ryzen 9 7950X CPU, respectively.

We choose common pretrained DNN **architectures** from the literature (Hacohen et al., 2022; Gupte et al., 2024). For image datasets, we employ a Vision Transformer (ViT) (Dosovitskiy et al., 2021)

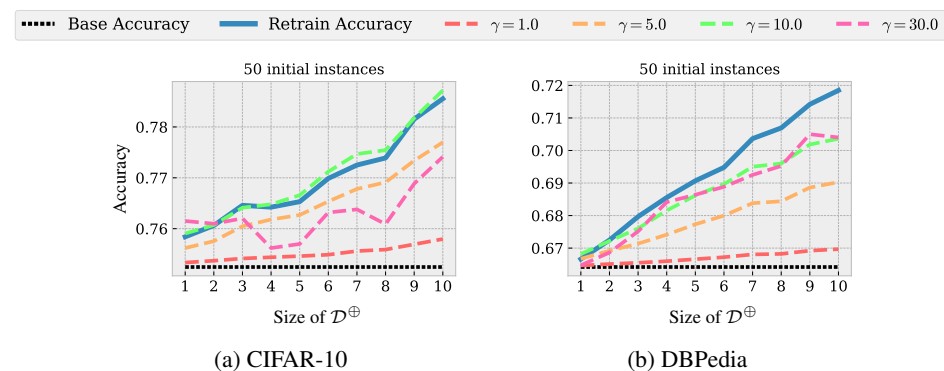

Figure 2: Accuracies after updating with different values for $\gamma$ in comparison to the baseline DNN and retraining.

with pretrained weights via self-supervised learning, complemented by a randomly initialized fully connected layer. Specifically, we use the DINOv2-ViT-S/14 model (Oquab et al., 2023) with a feature dimension of $D = 384$ in its final hidden layer. For text datasets, we employ the transformer-based pretrained language model BERT (Devlin et al., 2019). We utilize BERT-BASED-UNCASED from the Huggingface library (Wolf et al., 2020) with a feature dimension of $D = 768$ and a maximum sequence length of 512. We train each DNN by finetuning for 200 epochs, employing the Rectified Adam optimizer (Liu et al., 2019) with a training batch size of 64, a learning rate of 0.01 for images and 0.1 for text, and weight decay of 0.0001. In addition, we utilize a cosine annealing learning rate scheduler. These hyperparameters were determined empirically to be effective across all datasets by investigating the loss convergence on validation splits.

## 4.2 EXPERIMENTS

**Hyperparameter Ablation:** In equation 5, we introduced the hyperparameter $\gamma$, which controls the step size of our update. Intuitively, this factor determines the extent to which the DNN is influenced by the new dataset $\mathcal{D}^\oplus$. This factor is essential to control the update process and avoid issues such as catastrophic forgetting. Similarly, first-order and MC-based updates also utilize this factor to mitigate such problems. For further details, we refer to Appendix A.

To investigate the influence of $\gamma$ and determine a suitable value for all subsequent experiments, we conduct a simple ablation study on two datasets. The results of our update are shown here, while the results for first-order and MC-based updates can be found in Appendix B. We determine the value of $\gamma$ in this manner since an extensive hyperparameter search for update methods is typically impractical in an online setting (De Lange et al., 2021). Hence, fixing a value beforehand is necessary. We randomly sample an initial dataset $\mathcal{D}$ of 50 instances and train our baseline DNN. Subsequently,

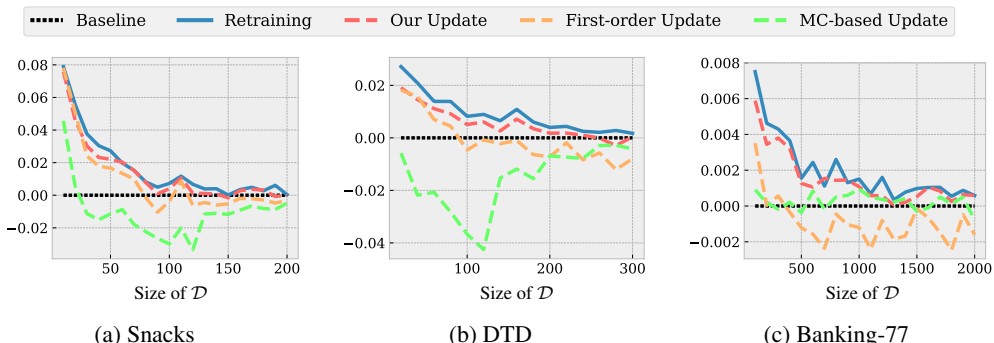

Figure 3: Accuracy improvement curves for six benchmark datasets, showing the difference in accuracy between retrained and updated DNNs for varying sizes of $\mathcal{D}$.

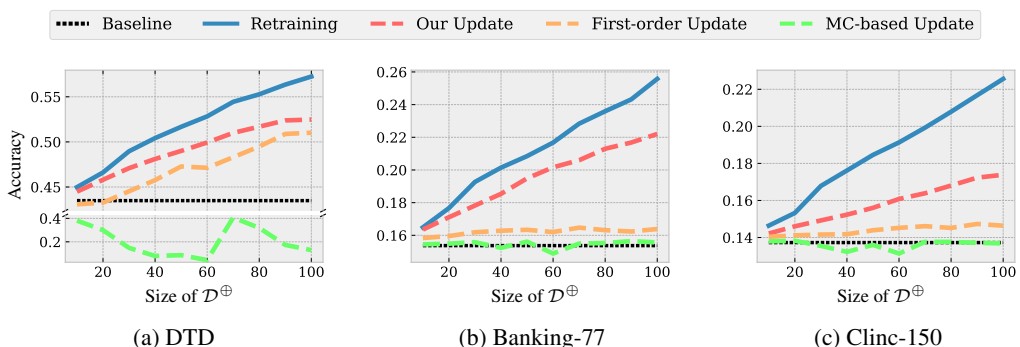

(a) DTD      (b) Banking-77      (c) Clinc-150

Figure 4: Accuracy curves for three benchmark datasets after updating and retraining DNNs for varying sizes of $\mathcal{D}^{\oplus}$.

updates and retraining are performed on randomly sampled datasets $|\mathcal{D}^{\oplus}| \in \{1, \ldots, 10\}$, and the accuracy is computed on a validation split. We repeat this process for different values of $\gamma$.

The resulting curves in Figure 2 indicate that our update with $\mathcal{D}^{\oplus}$ consistently achieves better performance than the baseline DNN that is only trained on $\mathcal{D}$. For both CIFAR-10 and DBPedia, updating with $\gamma = 1$ does not yield accuracies close to retraining, suggesting that the update is too weak. By increasing $\gamma$, we observe accuracies much closer to complete retraining, with $\gamma = 10$ being sufficient for CIFAR-10 and DBPedia. For CIFAR-10, we also notice that a very high value, i.e., $\gamma = 30$, can lead to worse performance, likely due to catastrophic forgetting. To ensure effective updates across all datasets, we will be using $\gamma = 10$ in all subsequent experiments. While this may not be optimal for some datasets, it should ensure a consistently working update in all cases.

**Different Learning Stages:** To investigate how our update behaves at different stages of learning, we train the baseline DNN on varying sizes of initial datasets $\mathcal{D}$ and update it with a new dataset of fixed size $|\mathcal{D}^{\oplus}| = 10$. To better visualize the differences, we report accuracy improvement of updated/retrained DNNs relative to the baseline in Figure 3. The results demonstrate that our updates provide the highest accuracy improvements across all datasets, highlighting the effective and consistent performance improvements of our update at different learning stages. While first-order and MC-based updates are also effective in earlier stages (when $|\mathcal{D}| < 50$), they tend to be less effective and even deteriorate accuracy in later stages. Compared to the first-order update, our update consistently enhances performance due to including the Hessian. As the Hessian considers curvature information about the posterior, the update is more robust regarding the choice of $\gamma$.

**Varying Size of $\mathcal{D}^{\oplus}$:** To investigate our update's behavior with an increasing number of new data points in $\mathcal{D}^{\oplus}$, we train a baseline DNN with a fixed initial dataset $|\mathcal{D}| = 100$ and vary the size of the new dataset $|\mathcal{D}^{\oplus}| \in \{10, 20, \ldots, 100\}$. We report the results for the most complex datasets DTD, Banking-77, and Clinc-150. In Figure 4, we observe that as the size of $\mathcal{D}^{\oplus}$ increases, the accuracy of retraining, our update, and the first-order update consistently improves. In contrast, MC-based updates result in worse accuracies than the baseline, indicating that it is not suited for an increasing size of $\mathcal{D}^{\oplus}$. Considering our update, we see that it consistently achieves better accuracies compared to competitors, regardless of the complexity of the dataset. Moreover, first-order updates seem to be less effective on the more complex datasets such as Banking-77 and Clinc-150, highlighting the importance of the Hessian.

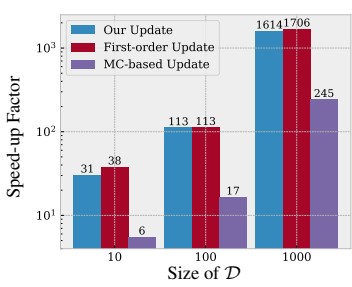

Figure 5: Speed-up of update methods compared to retraining.

**Time Comparison:** Finally, to evaluate the speed of updates, we fix the size of the new dataset to $|\mathcal{D}^{\oplus}| = 10$ and compute the speed-up relative to retraining by varying the initial dataset size $|\mathcal{D}|$. Figure 5 presents the speed-up factors on CIFAR-10. All update methods are faster than retraining, with the first-order update being the fastest. For example, with an initial dataset of $|\mathcal{D}| = 1000$, the

first-order update is about 1700 times faster than retraining. Notably, our update provides a similar speed-up factor while yielding more effective updates by using the closed-form Hessian update. Compared to MC-based updates, both the first-order and our update are significantly faster.

# 5 DEEP ACTIVE LEARNING

In this section, we examine the proposed update in AL. First, we introduce a new framework that uses our updates to exploit label information during batch construction. Essentially, this approach mimics single instance AL, in which the model is retrained after each label acquisition. Next, we employ our update to approximate an optimal look-ahead strategy. Instead of obtaining future performance of the DNN with expensive retraining, we realize this through our update. Here, we average metrics over 30 repetitions to account for reproducibility challenges in AL (Munjal et al., 2022). Labeling budgets and acquisition sizes differ based on the complexity of a dataset. A more detailed experimental setup and all learning curves, including ones that report absolute values, are available in Appendix F.

## 5.1 IMPROVED BATCH SELECTION VIA UPDATES

A naive and suboptimal way of using sequential selection strategies for batch selection is to use the top-$b$ scoring instances (Kirsch et al., 2023). Our idea is to overcome the necessity of batch strategies by using the proposed update method with sequential strategies as a fast alternative to retraining. Thus, we iteratively select the highest-scoring instance $b$ times and update the DNN between each selection. After acquiring $b$ labels, we retrain the DNN similar to batch selection strategies. An algorithm implementing this idea can be found in Appendix F.

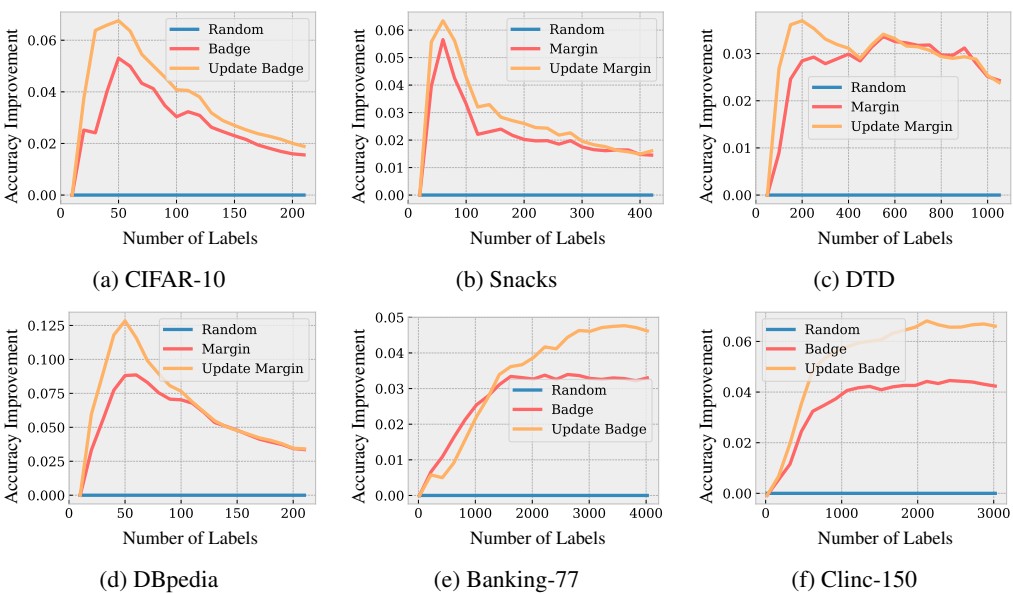

Figure 6: Accuracy improvement curves for different strategies and datasets showing the accuracy difference between the respective selection strategy and random instance selection.

The hypothesis is that already well-performing sequential selection strategies (Ren et al., 2021) can simply be used in a batch setting and that our framework can achieve higher performance compared to selecting the top-$b$ instances. Here, we consider the widely used strategy Margin, which has proven to be effective in several studies (Bahri et al., 2022; Huseljic et al., 2021). Additionally, we are interested in whether this idea can also act as a replacement for the diversity component of a batch selection strategy. Therefore, we also evaluate the popular strategy Badge (Ash et al., 2020) in combination with our updates.

Figure 6 shows the accuracy improvement curves relative to a random instance selection. The results confirm our hypothesis. The query strategies using our updates outperform the respective top-$b$

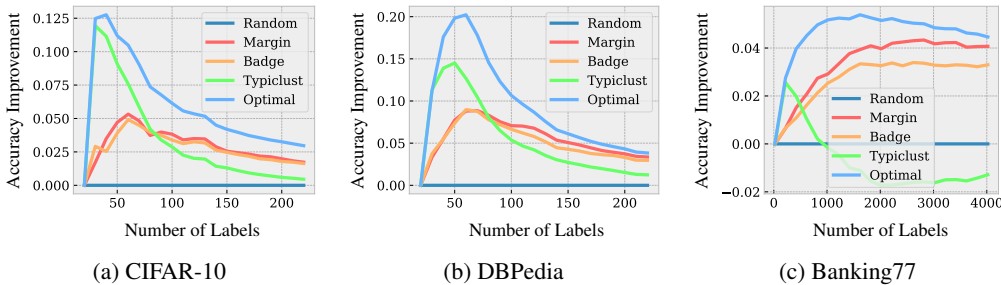

(a) CIFAR-10       (b) DBPedia       (c) Banking77

Figure 7: Accuracy improvement over random selection of popular selection strategies compared to our upper baseline approximating optimal batch selection.

selection strategies. Specifically, we see improved performance in early stages when redundancy within a batch plays an important role. Moreover, combining our update with Badge also results in improved accuracy. This indicates that *selecting a single instance and updating the DNN* leads to a more effective selection than using the $k$-MEANS++ algorithm as proposed in Badge.

## 5.2 UPDATING IN LOOK-AHEAD STRATEGIES

The idea of look-ahead strategies is to select instances that, once labeled and added to the labeled pool, maximize the performance of the model (Roy & McCallum, 2001). Unlike uncertainty- or diversity-based approaches, look-ahead strategies select instances based on an optimal criterion: the model's actual performance. However, they are often neglected in deep AL due to the high computational requirements. One of the biggest bottlenecks in the selection is retraining. DNNs are not well-suited for this due to their long training process. For this reason, we employ our proposed update to make this feasible.

Here, we consider a near-optimal strategy with access to ground truth information, including labels and validation datasets. It can be considered as an upper baseline in deep AL research. For the selection, we randomly sample 2000 subsets, each with a size equal to the acquisition size, and assess how their addition to the labeled pool affects the performance. The batch leading to the highest performance gain is selected. While this approach would traditionally require 2000 times of retraining–making it infeasible with DNNs–our update enables the efficient use of this strategy.

In this experiment, we also include the recently proposed Typiclust strategy, which has demonstrated strong performance (Hacohen et al., 2022), especially in early stages of AL. Figure 7 presents the resulting accuracy improvement curves compared to random instance selection. Our optimal strategy, using updates rather than full retraining, performs exceptionally well, consistently outperforming all competitors. Based on these results, we can see that the current selection strategies still have much potential for improvement. Interestingly, we can see that Typiclust's selection in the early stages of AL seems to be close to an optimal selection but declines in effectiveness in later stages.

## 6 CONCLUSION

In this article, we proposed an efficient second-order update for DNNs in AL using the Gaussian posterior of a last-layer LA. It achieves low computational complexity through a closed-form computation of the required inverse Hessian. An extensive experimental evaluation showed that the proposed update provides an efficient alternative to retraining. Based on this observation, we introduced a new batch selection framework by sequentially updating the DNN after each label acquisition, offering a new perspective on constructing batches without resorting to heuristics such as clustering. Furthermore, we realized a look-ahead strategy as a feasible upper baseline approximating optimal batch selection, highlighting the great potential for improvement in current research on batch selection strategies. In future work, we plan to utilize the proposed updates to enhance look-ahead selection strategies (Roy & McCallum, 2001) in deep AL. As these strategies are based on decision-theoretic principles, they naturally balance explorative and exploitative instance selection, a key challenge in AL (Li et al., 2024).

## REPRODUCIBILITY STATEMENT

Reproducibility is an essential factor in active learning. To ensure reproducibility, we averaged metrics over repeated experiments. We opted not to report standard errors to maintain clarity in visualizations, especially since standard errors were negligible and added little information. Each experiment evaluating updates in Section 4 was repeated 10 times, while we increased this number to 30 for all active learning experiments in Section 5. The code and detailed instructions for setting up and running the experiments are available in a GitHub repository, ensuring that our work can be easily reproduced and built upon.

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

## A    RELATED BAYESIAN UPDATE METHODS

We formally introduce Bayesian update methods used for comparison in the main part, including Monte Carlo (MC)-based updates and our method's simpler variant, first-order updates.

### A.1    MC-BASED BAYESIAN UPDATES

MC-based Bayesian neural networks (BNNs) such as deep ensembles and MC-Dropout draw samples, or hypotheses, from an (approximate) posterior distribution $q(\boldsymbol{\omega}|\mathcal{D})$. To obtain these samples, deep ensembles train multiple randomly initialized deep neural networks (DNNs), while MC-Dropout randomly sets a portion of parameters to zero for multiple inference steps. We refer to (Gawlikowski et al., 2023) for an in-depth explanation of these techniques.

As MC-based BNNs only have access to these samples, the idea behind the MC-based update method is to weigh these samples. More specifically, MC-based updates, as used in (Tan et al., 2021), assume a priori that every hypothesis $\boldsymbol{\omega}_1, \ldots, \boldsymbol{\omega}_M \sim q(\boldsymbol{\omega}|\mathcal{D})$ is equally likely to explain the new dataset $\mathcal{D}^{\oplus}$. Hence, the approximate distribution *over the drawn members* can be defined as a categorical distribution[3] with parameters $\hat{\boldsymbol{p}} = (\hat{p}_1, \ldots, \hat{p}_M)^{\mathrm{T}}$:

$$q(\boldsymbol{\omega}_m|\mathcal{D}) = \mathrm{Cat}(m|\hat{\boldsymbol{p}}) = \hat{p}_m = 1/M, \tag{7}$$

where $M$ is the number of drawn ensemble members. The updated posterior distribution, which includes the new dataset $\mathcal{D}^{\oplus}$, is computed through Bayes' theorem:

$$q(\boldsymbol{\omega}_m|\mathcal{D}^{\oplus}, \mathcal{D}) = \mathrm{Cat}(m|\hat{\boldsymbol{p}}^{\mathrm{upd}}) \propto q(\boldsymbol{\omega}_m|\mathcal{D}) \prod_{(\boldsymbol{x},y)\in\mathcal{D}^{\oplus}} p(y|\boldsymbol{x}, \boldsymbol{\omega}_m) \tag{8}$$

$$= \hat{p}_m \prod_{(\boldsymbol{x},y)\in\mathcal{D}^{\oplus}} [\mathrm{softmax}(f^{\boldsymbol{\omega}_m}(\boldsymbol{x}))]_y = \hat{z}_m, \tag{9}$$

which is a categorical distribution with parameters $\hat{\boldsymbol{p}}^{\mathrm{upd}} = (\hat{p}_1^{\mathrm{upd}}, \ldots, \hat{p}_M^{\mathrm{upd}})^{\mathrm{T}}$ that we obtain after normalizing $\hat{\boldsymbol{z}} = (\hat{z}_1, \ldots, \hat{z}_M)^{\mathrm{T}}$. Intuitively, the importance of each hypothesis is determined by its likelihood of explaining the new dataset $\mathcal{D}^{\oplus}$. This approximation $q(\boldsymbol{\omega}_m|\mathcal{D}^{\oplus}, \mathcal{D})$ of the posterior distribution allows us to make new predictions by evaluating the predictive distribution from equation 1 accordingly:

$$p(y|\boldsymbol{x}, \mathcal{D}^{\oplus}, \mathcal{D}) = \mathbb{E}_{q(\boldsymbol{\omega}|\mathcal{D}^{\oplus}, \mathcal{D})}[p(y|\boldsymbol{x}, \boldsymbol{\omega})] \approx \sum_{m=1}^{M} p(y|\boldsymbol{x}, \boldsymbol{\omega}_m) \cdot q(\boldsymbol{\omega}_m|\mathcal{D}^{\oplus}, \mathcal{D}) \tag{10}$$

$$= \sum_{m=1}^{M} [\mathrm{softmax}(f^{\boldsymbol{\omega}_m}(\boldsymbol{x}))]_y \cdot \hat{p}_m^{\mathrm{upd}}, \tag{11}$$

which is a weighted average of the sampled hypotheses. Employing the update from equation 9 may lead to catastrophic forgetting. Thus, to control the influence of the new dataset $\mathcal{D}^{\oplus}$, we introduce the hyperparameter $\gamma$:

$$q(\boldsymbol{\omega}_m|\mathcal{D}^{\oplus}, \mathcal{D}) \propto \hat{p}_m \left( \prod_{(\boldsymbol{x},y)\in\mathcal{D}^{\oplus}} [\mathrm{softmax}(f^{\boldsymbol{\omega}_m}(\boldsymbol{x}))]_y \right)^{\gamma}. \tag{12}$$

### A.2    FIRST-ORDER BAYESIAN UPDATES

Our proposed update method from the main text approximates the new posterior $q(\boldsymbol{\omega}|\mathcal{D}, \mathcal{D}^{\oplus})$ by applying an optimization step via Gauss-Newton and estimating the new covariance. For comparison, we evaluate an update based on a first-order optimization step, leading to a less complex and faster method. This also allows us to ablate the importance of the Hessian.

Assume a binary classification problem where we have an approximate posterior distribution $q(\boldsymbol{\omega}|\mathcal{D}) = \mathcal{N}(\boldsymbol{\omega}|\hat{\boldsymbol{\mu}}, \hat{\boldsymbol{\Sigma}})$ from an LA and observe the new dataset $\mathcal{D}^{\oplus}$. Our goal is to update the

---

[3]Equivalently, one can define the approximate distribution via multiple Dirac deltas.

approximate posterior distribution $q(\boldsymbol{\omega}|\mathcal{D})$ by considering it as the new prior distribution:

$$q(\boldsymbol{\omega}|\mathcal{D}, \mathcal{D}^{\oplus}) = \mathcal{N}(\boldsymbol{\omega}|\hat{\boldsymbol{\mu}}^{\text{upd}}, \hat{\boldsymbol{\Sigma}}^{\text{upd}}) \overset{\propto}{\sim} q(\boldsymbol{\omega}|\mathcal{D}) \prod_{(\boldsymbol{x},y)\in\mathcal{D}^{\oplus}} p(y|\boldsymbol{x}, \boldsymbol{\omega}). \tag{13}$$

The first-order update is defined by using a gradient optimization step to obtain an updated mean:

$$\hat{\boldsymbol{\mu}}^{\text{upd}} = \hat{\boldsymbol{\mu}} - \gamma \left( \sum_{(\boldsymbol{x},y)\in\mathcal{D}^{\oplus}} \left( \sigma(\boldsymbol{h}^{\text{T}}\hat{\boldsymbol{\mu}}) - y \right) \boldsymbol{h_x} \right), \tag{14}$$

where $\gamma$ is the step size (or learning rate) introduced to avoid catastrophic forgetting. The covariance matrix must only be recomputed at the update's end if meaningful uncertainty estimates are of interest.

The updating strategy in equation 14 is equivalent to the continual learning strategy from Ritter et al. (2018a). There, the updated posterior distribution is defined as

$$\log p(\boldsymbol{\omega}|\mathcal{D}, \mathcal{D}^{\oplus}) \propto \sum_{(\boldsymbol{x},y)\in\mathcal{D}^{\oplus}} \log p(y|\boldsymbol{x}, \boldsymbol{\omega}) - \frac{1}{2}(\boldsymbol{\omega} - \hat{\boldsymbol{\mu}})^T \hat{\boldsymbol{\Sigma}}^{-1}(\boldsymbol{\omega} - \hat{\boldsymbol{\mu}}), \tag{15}$$

where the second term is the prior $q(\boldsymbol{\omega}|\mathcal{D})$, penalizing large deviations of $\boldsymbol{\omega}$ from the prior mean $\hat{\boldsymbol{\mu}}$. The updated mean is then given by

$$\hat{\boldsymbol{\mu}}^{\text{upd}} = \arg\max_{\boldsymbol{\omega}} \sum_{(\boldsymbol{x},y)\in\mathcal{D}^{\oplus}} \log p(y|\boldsymbol{x}, \boldsymbol{\omega}) - \frac{1}{2}(\boldsymbol{\omega} - \hat{\boldsymbol{\mu}})^T \hat{\boldsymbol{\Sigma}}^{-1}(\boldsymbol{\omega} - \hat{\boldsymbol{\mu}}). \tag{16}$$

Ritter et al. (2018a) use multiple optimization steps via gradient descent to obtain the new mean $\hat{\boldsymbol{\mu}}^{\text{upd}}$. Since we defined an update as a single optimization step and we start with $\boldsymbol{\omega} = \hat{\boldsymbol{\mu}}$, the regularization term evaluates to zero. Thus, the updated mean is obtained through a gradient step on the new data likelihood, simplifying the optimization to equation 14.

## B HYPERPARAMETER ABLATION

For all introduced Bayesian update methods, the hyperparameter $\gamma$ controls the influence of the new dataset $\mathcal{D}^{\oplus}$ on the posterior distribution $q(\boldsymbol{\omega}|\mathcal{D})$. In Section 4, we conducted a small ablation study, similar to a hyperparameter search, to determine an appropriate value for $\gamma$. We intentionally did not search a optimal value of $\gamma$ for every dataset since extensive hyperparameter search for update methods is impractical in an online setting (De Lange et al., 2021). Using the same setup as in Section 4, we consider the CIFAR-10 and DBPedia datasets. Our baseline DNN is trained on a randomly sampled initial dataset $\mathcal{D}$ of 50 instances. We then update and retrain the DNN with varying sizes of the new dataset $|\mathcal{D}^{\oplus}| \in \{1, \ldots, 10\}$.

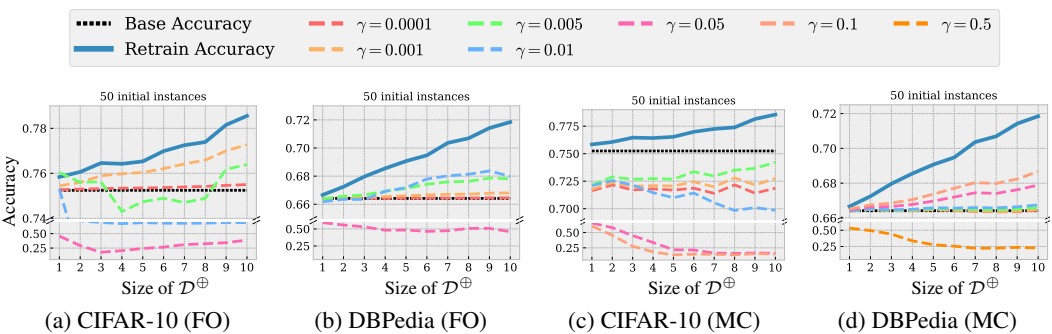

(a) CIFAR-10 (FO)     (b) DBPedia (FO)     (c) CIFAR-10 (MC)     (d) DBPedia (MC)

Figure 8: Accuracies after updating with different values for $\gamma$ in comparison to the baseline DNN and retraining.

Considering the first-order update, we see that not all values of $\gamma$ improve accuracy. Especially, high values can cause a collapse in accuracy, likely due to catastrophic forgetting. We find that

updates with $\gamma = 0.001$ and $\gamma = 0.01$ perform best. Since $\gamma = 0.001$ generates the highest accuracy for CIFAR-10 and does not lead to a worse performance for DBPedia, we use it for the remaining experiments. Considering MC-based update, we see that no value of gamma provides an improvement in accuracy on CIFAR-10. In contrast, for DBPedia we see that $\gamma = 0.1$ improve accuracy the most. Considering the results across datasets, we select $\gamma = 0.005$ for images and $\gamma = 0.01$ for text.

## C MULTI-CLASS LAST-LAYER UPDATE

Here, we outline three different options for performing the update step for LA in a multi-class setting with $K > 2$. In a binary setting, a last-layer LA uses a parameter vector $\boldsymbol{\omega} \in \mathbb{R}^D$. However, in a multi-class setting, we have a parameter vector $\boldsymbol{\omega}_y$ for each class $y \in \mathcal{Y}$. In the literature, several approximations haven been proposed to model these parameter vectors' distribution.

The most complex approximation would be to concatenate these vectors and model them through a multi-variate normal distribution:

$$q\left(\boldsymbol{\omega} \mid \mathcal{D}\right) = \mathcal{N}(\boldsymbol{\omega}|\hat{\boldsymbol{\mu}}, \hat{\boldsymbol{\Sigma}}) \text{ with } \boldsymbol{\omega}, \hat{\boldsymbol{\mu}} \in \mathbb{R}^{K \cdot D}, \hat{\boldsymbol{\Sigma}} \in \mathbb{R}^{(K \cdot D) \times (K \cdot D)}. \tag{17}$$

This approximation can estimate a covariance between each pair of parameters. However, this expressiveness comes at the cost of a large covariance matrix $\hat{\boldsymbol{\Sigma}}$ to be estimated. This is particularly costly for many classes $K$ combined with a high feature dimension $D$ from a DNN, such that corresponding updates via second-order optimization would no longer be efficient.

Spiegelhalter & Lauritzen (1990) presented a more efficient approximation in which the class-wise parameter vectors are arranged column-wisely as a matrix. Their joint distribution is then modeled through a matrix normal distribution:

$$q\left(\boldsymbol{\omega}|\mathcal{D}\right) = \mathcal{MN}(\boldsymbol{\omega}|\hat{\boldsymbol{\mu}}, \hat{\boldsymbol{\Gamma}}, \hat{\boldsymbol{\Sigma}}) \text{ with } \boldsymbol{\omega}, \hat{\boldsymbol{\mu}} \in \mathbb{R}^{D \times K}, \hat{\boldsymbol{\Sigma}} \in \mathbb{R}^{D \times D}, \hat{\boldsymbol{\Gamma}} \in \mathbb{R}^{K \times K}. \tag{18}$$

This approximation captures the covariance between each pair of parameter vectors via the matrix $\hat{\boldsymbol{\Gamma}}$ and each pair of hidden features via the matrix $\hat{\boldsymbol{\Sigma}}$. Both matrices have to be iteratively recomputed while updating. We refer to (Spiegelhalter & Lauritzen, 1990) for more details.

Liu et al. (2020) presented an even faster approximation and showed its effectiveness in combination with an LA for supervised learning. Therefore, we employ this approximation in the multi-class setting. The idea is to determine an upper-bound covariance matrix shared by all class-wise parameter vectors in their respective multivariate normal distribution, which is then defined for class $y \in \mathcal{Y}$ as:

$$q(\boldsymbol{\omega}_y \mid \mathcal{D}) = \mathcal{N}(\hat{\boldsymbol{\mu}}_y|\hat{\boldsymbol{\Sigma}}) \text{ with } \boldsymbol{\omega}_y, \hat{\boldsymbol{\mu}}_y \in \mathbb{R}^D, \hat{\boldsymbol{\Sigma}} \in \mathbb{R}^{D \times D}. \tag{19}$$

The upper-bound covariance matrix $\hat{\boldsymbol{\Sigma}} \in \mathbb{R}^{D \times D}$ corresponds to the inverse Hessian of the negative log posterior evaluated at the MAP estimate $\hat{\boldsymbol{\mu}}$ given training data $\mathcal{D}$ and a prior covariance matrix $\boldsymbol{I}$. It is given by

$$\hat{\boldsymbol{\Sigma}} = \boldsymbol{H}^{-1}, \text{ where } \quad \boldsymbol{H} = \sum_{(\boldsymbol{x},y) \in \mathcal{D}} p_{\boldsymbol{x}}^{\star}(1 - p_{\boldsymbol{x}}^{\star})\boldsymbol{h}_{\boldsymbol{x}}\boldsymbol{h}_{\boldsymbol{x}}^T + \boldsymbol{I}, \tag{20}$$

where $p_{\boldsymbol{x}}^{\star} = \max_y p(y|\boldsymbol{x}, \boldsymbol{\omega})$ is the maximum probability outputted by the the DNN. An even simpler approximation is to assume a Gaussian likelihood for the covariance, leading to the following formulation

$$\hat{\boldsymbol{\Sigma}} = \boldsymbol{H}^{-1}, \text{ where } \quad \boldsymbol{H} = \sum_{(\boldsymbol{x},y) \in \mathcal{D}} \boldsymbol{h}_{\boldsymbol{x}}\boldsymbol{h}_{\boldsymbol{x}}^T + \boldsymbol{I}. \tag{21}$$

which has been empirically shown to work more robustly when estimating uncertainties (Liu et al., 2023). For this reason and its computational efficiency, we employ this approximation in our method.

Analog to the updates for binary classification in equation 5, we implement the updates of the mean parameter vector for class $y \in \mathcal{Y}$ and the covariance matrix $\hat{\boldsymbol{\Sigma}} \in \mathbb{R}^{D \times D}$ shared across the classes $\mathcal{Y}$ given a new dataset $\mathcal{D}^{\oplus}$ based on the Gauss-Newton algorithm leading to:

$$\hat{\boldsymbol{\mu}}_y^{\text{upd}} = \hat{\boldsymbol{\mu}} - \boldsymbol{H}^{-1}(\hat{\boldsymbol{\mu}}, \hat{\boldsymbol{\Sigma}}, \mathcal{D}^{\oplus}) \sum_{(\boldsymbol{x},y') \in \mathcal{D}^{\oplus}} (p_{\boldsymbol{x}}^{\star} - \delta(y = y'))\boldsymbol{h}_{\boldsymbol{x}}, \tag{22}$$

$$\hat{\boldsymbol{\Sigma}}^{\text{upd}} = \boldsymbol{H}^{-1}(\hat{\boldsymbol{\mu}}^{\text{upd}}, \hat{\boldsymbol{\Sigma}}, \mathcal{D}^{\oplus}), \tag{23}$$

where $\delta(\cdot)$ is the Dirac delta function. We efficiently compute the updated inverse Hessian using the Woodbury identity as presented in the upcoming Appendix D.

## D    EFFICIENT HESSIAN INVERSION VIA WOODBURY IDENTITY

Here, we provide the derivation of the update from Eq. equation 6, which uses the Woodbury (matrix) identity (Woodbury, 1950) for efficient inversion of the Hessian during updates. Assume we employed an LA and have the current approximate posterior distribution $q(\boldsymbol{\omega}|\mathcal{D}) = \mathcal{N}(\boldsymbol{\omega}|\hat{\boldsymbol{\mu}}, \hat{\boldsymbol{\Sigma}})$. To incorporate the information of the new dataset $\mathcal{D}^{\oplus}$ into the LA's current covariance $\hat{\boldsymbol{\Sigma}}$, we first need to calculate the Hessian of the new negative log posterior $q(\boldsymbol{\omega}|\mathcal{D}, \mathcal{D}^{\oplus})$ with respect to $\boldsymbol{\omega}$. The Hessian of the new negative log posterior $-q(\boldsymbol{\omega}|\mathcal{D}, \mathcal{D}^{\oplus})$ is given by

$$\boldsymbol{H} = -\nabla_{\boldsymbol{\omega}}^2 \log q(\boldsymbol{\omega}|\mathcal{D}, \mathcal{D}^{\oplus}) = \boldsymbol{I} + \sum_{(\boldsymbol{x},y)\in\mathcal{D}} p_{\boldsymbol{x}}(1-p_{\boldsymbol{x}})\boldsymbol{h}_{\boldsymbol{x}}\boldsymbol{h}_{\boldsymbol{x}}^T + \sum_{(\boldsymbol{x},y)\in\mathcal{D}^{\oplus}} p_{\boldsymbol{x}}(1-p_{\boldsymbol{x}})\boldsymbol{h}_{\boldsymbol{x}}\boldsymbol{h}_{\boldsymbol{x}}^T, \tag{24}$$

$$= \hat{\boldsymbol{\Sigma}}^{-1} + \sum_{(\boldsymbol{x},y)\in\mathcal{D}^{\oplus}} p_{\boldsymbol{x}}(1-p_{\boldsymbol{x}})\boldsymbol{h}_{\boldsymbol{x}}\boldsymbol{h}_{\boldsymbol{x}}^T, \tag{25}$$

where we see the updated Hessian is given by a sum of the old negative log posterior's precision matrix with the precision matrix of the new dataset $\mathcal{D}^{\oplus}$. Note, however, that we require the *inverse* Hessian $\boldsymbol{H}^{-1}$ for both the Gaussian posterior in the LA and the optimization step via Gauss-Newton. Depending on the size of $\mathcal{D}^{\oplus}$ and the assumed likelihood for the Hessian, this computation can significantly slow down the efficiency of our update since we must compute an inverse with each new incoming batch of instances. Thus, to ensure efficient updates, we employ the Woodbury identity, which is given in a simplified form by

$$(\boldsymbol{A} + \boldsymbol{u}\boldsymbol{v}^T)^{-1} = \boldsymbol{A}^{-1} - \frac{\boldsymbol{A}^{-1}\boldsymbol{u}\boldsymbol{v}^T\boldsymbol{A}^{-1}}{1 + \boldsymbol{v}^T\boldsymbol{A}^{-1}\boldsymbol{u}}, \tag{26}$$

and allows us to obtain an updated inverse Hessian directly, avoiding calculation of the inverse of the updated precision matrix. Setting $\boldsymbol{u} = p_{\boldsymbol{x}}(1-p_{\boldsymbol{x}})\boldsymbol{h}_{\boldsymbol{x}}$ and $\boldsymbol{v} = \boldsymbol{h}_{\boldsymbol{x}}$, we obtain:

$$\boldsymbol{H}^{-1} = \left(\hat{\boldsymbol{\Sigma}}^{-1} + p_{\boldsymbol{x}}(1-p_{\boldsymbol{x}})\boldsymbol{h}_{\boldsymbol{x}}\boldsymbol{h}_{\boldsymbol{x}}^T\right)^{-1} = \hat{\boldsymbol{\Sigma}} - \frac{p_{\boldsymbol{x}}(1-p_{\boldsymbol{x}})}{1 + \boldsymbol{h}_{\boldsymbol{x}}\hat{\boldsymbol{\Sigma}}\boldsymbol{h}_{\boldsymbol{x}} \cdot p_{\boldsymbol{x}}(1-p_{\boldsymbol{x}})}\hat{\boldsymbol{\Sigma}}\boldsymbol{h}_{\boldsymbol{x}}\boldsymbol{h}_{\boldsymbol{x}}\hat{\boldsymbol{\Sigma}}, \tag{27}$$

which is the update from equation 6. In the multi-class setting, we employ the covariance matrix of equation 21 by assuming a Gaussian likelihood leading to the following inverse Hessian:

$$\boldsymbol{H}^{-1} = \left(\hat{\boldsymbol{\Sigma}}^{-1} + \boldsymbol{h}_{\boldsymbol{x}}\boldsymbol{h}_{\boldsymbol{x}}^T\right)^{-1} = \hat{\boldsymbol{\Sigma}} - \frac{\hat{\boldsymbol{\Sigma}}\boldsymbol{h}_{\boldsymbol{x}}\boldsymbol{h}_{\boldsymbol{x}}\hat{\boldsymbol{\Sigma}}}{1 + \boldsymbol{h}_{\boldsymbol{x}}\hat{\boldsymbol{\Sigma}}\boldsymbol{h}_{\boldsymbol{x}}}, \tag{28}$$

where we set $\boldsymbol{u} = \boldsymbol{v} = \boldsymbol{h}_{\boldsymbol{x}}$.

## E    SUMMARY OF DATASETS

**CIFAR** (Krizhevsky, 2009) consists of 60,000 colored images with a low resolution of $32 \times 32$. There is a predetermined split of 50,000 images as training instances and 10,000 images as test instances. One variant of this dataset, **CIFAR-10**, is a coarse-grained task with ten broad classes such as automobile, dog, airplane, and ship. The classes are mutually exclusive meaning there is no overlap between, e.g., automobiles and trucks. **Snacks** (Matthijs, 2021) contains images of 20 different classes of snack foods. The 6,745 available images are split into 3,840 images for training, 955 for validation, and 920 for testing. Each image is 256 pixels wide while its height varies from 256 to 873 pixels. **DTD** (Cimpoi et al., 2014) is a texture-focused dataset that consists of 5,640 images divided into 47 different classes. There are 120 images for each class and image sizes range between 300x300 and 640x640. There is a predetermined split into three equally sized datasets for training, validation, and testing. **DBPedia** (Auer et al., 2007) is a larger text dataset with a medium class cardinality of 14 different classes. There are 14 different ontology-based classes. It consists of 560,000 training instances and 5,000 test instances. **Banking-77** (Casanueva et al., 2020) comes with a more complex task of conversational language understanding alongside intent detection. The combination of its high-class cardinality and a small pool of just 10,000 instances makes this data set even more challenging. **Clinc-150** (Larson et al., 2019) includes queries that are out-of-scope, i.e., queries that do not fall into any of the system's supported intents. It contains 150 in-scope intent classes, each with 100 train, 20 validation, and 30 test instances. Additionally, there are 100 train and validation out-of-scope instances, and 1,000 out-of-scope test instances.

# F   ACTIVE LEARNING: UPDATES FOR BATCH SELECTION AND LOOK-AHEAD

**Algorithm:** The main idea from Section 5 is to transform any sequential selection strategy into a batch strategy by using the proposed update method as a fast alternative to retraining. Typically, when using a sequential selection strategy, the naive idea in a batch setting is to select the instances that resulted in the top-$b$ highest scores, where the score is defined by the respective selection strategy. Our method can now update the DNN after each acquired label instead of selecting the top-$b$ instances. Hence, we have a similar scenario as if we would perform single-instance acquisitions but avoid the retraining after each acquired label. After aggregating a batch of $b$ instances, we retrain the DNN.

Algorithm 1 summarizes this idea. Given a standard AL setting with labeled data $\mathcal{L}$ and unlabeled data $\mathcal{U}$, we first calculate the approximate posterior distribution $q(\boldsymbol{\omega}|\mathcal{L})$ via LA. Subsequently, we iteratively select a new instance $\boldsymbol{x}$ and acquire its label $y$ based on a given selection strategy $\alpha(\cdot)$ and update the approximate posterior distribution $q(\boldsymbol{\omega}|\{(\boldsymbol{x},y)\},\mathcal{L})$ according to equation 5. We repeat this procedure for $b$ acquisitions.

---

**Algorithm 1** Updating in AL

---

**Require:** Labeled data $\mathcal{L}$, unlabeled data $\mathcal{U}$, selection strategy $\alpha$, acquisition size $b$, BNN $q(\boldsymbol{\omega}|\mathcal{L})$
1: New acquisitions $\mathcal{D}^{\oplus} = \{\}$
2: **for** $i = 1, \ldots, b$ **do**
3:     Acquire next instance $\hat{\boldsymbol{x}} = \text{argmax}_{\boldsymbol{x} \in \mathcal{U}} \alpha(\boldsymbol{x}, \mathcal{U}, q(\boldsymbol{\omega}|\mathcal{D}^{\oplus}, \mathcal{L}))$
4:     Obtain new label $\hat{y}$ for instance $\hat{\boldsymbol{x}}$
5:     Extend new acquisitions $\mathcal{D}^{\oplus} \leftarrow \mathcal{D}^{\oplus} \cup \{(\hat{\boldsymbol{x}}, \hat{y})\}$
6:     Remove acquisition from $\mathcal{U} \leftarrow \mathcal{U} \setminus \hat{\boldsymbol{x}}$
7:     Update posterior distribution $q(\boldsymbol{\omega}|\mathcal{D}^{\oplus}, \mathcal{L})$
8: **end for**
9: **return** Batch of new acquisitions $\mathcal{D}^{\oplus}$

---

**Experiments:** For the AL experiments in Section 5, we follow the recent work of (Hacohen et al., 2022; Gupte et al., 2024). We use all datasets (cf. Table 1) and initialize the labeled pool $\mathcal{L}$ with $b$ randomly sampled instances. In each AL cycle, we select $b$ new instances for labeling and fix the number of AL cycles to 20, resulting in a total budget of $B = 20 \cdot b$. The size $b$ is determined based on the dataset's complexity, ensuring that learning curves from random instance selection converge. Each AL selection strategy selected from 1,000 randomly sampled instances from the unlabeled pool $\mathcal{U}$ to speed up the selection process, except for Typiclust, where we increased the number to $10,000$ to avoid errors of $k$-MEANS. We employ the same architecture and training hyperparameters as described in the experimental setup in Section 4.1.

**Improved Batch Selection:** For the sequential selection strategy Margin (Bahri et al., 2022), we select batches of instances by following Algorithm 1 instead of the top-$b$ selection. In the case of Badge (Ash et al., 2020), we replace the $k$-MEANS++ algorithm that is typically used. Consequently, we select the instance with the highest gradient norm. All accuracy improvement curves are shown in Figure 9. Associated learning curves reporting the absolute accuracy are shown in Figure 10.

**Approximating Optimal Batch Selection:** In Section 5, we employed our update to approximate an optimal look-ahead batch selection strategy. The selection works as follows: We begin with a randomly initialized labeled pool. During selection, we deliberately avoid clustering schemes to simplify batch selection into picking a single instance per cluster. We argue that this enforced diversity can lead to suboptimal selection, especially in the later stages of active learning, where exploitation is more critical. Instead, we randomly sampled 2,000 batches, each matching the acquisition size. These batches likely include both diverse and non-diverse sets, allowing for exploration in the beginning and potential exploitation in the end. Each batch is used to update the model with our proposed method as a proxy for retraining. We then evaluated the performance of each updated model on a validation set. Finally, we selected the batch that yielded the highest performance improvement. The remaining accuracy improvement curves and learning curves reporting absolute values are shown in Figure 11 and Figure 12, respectively.

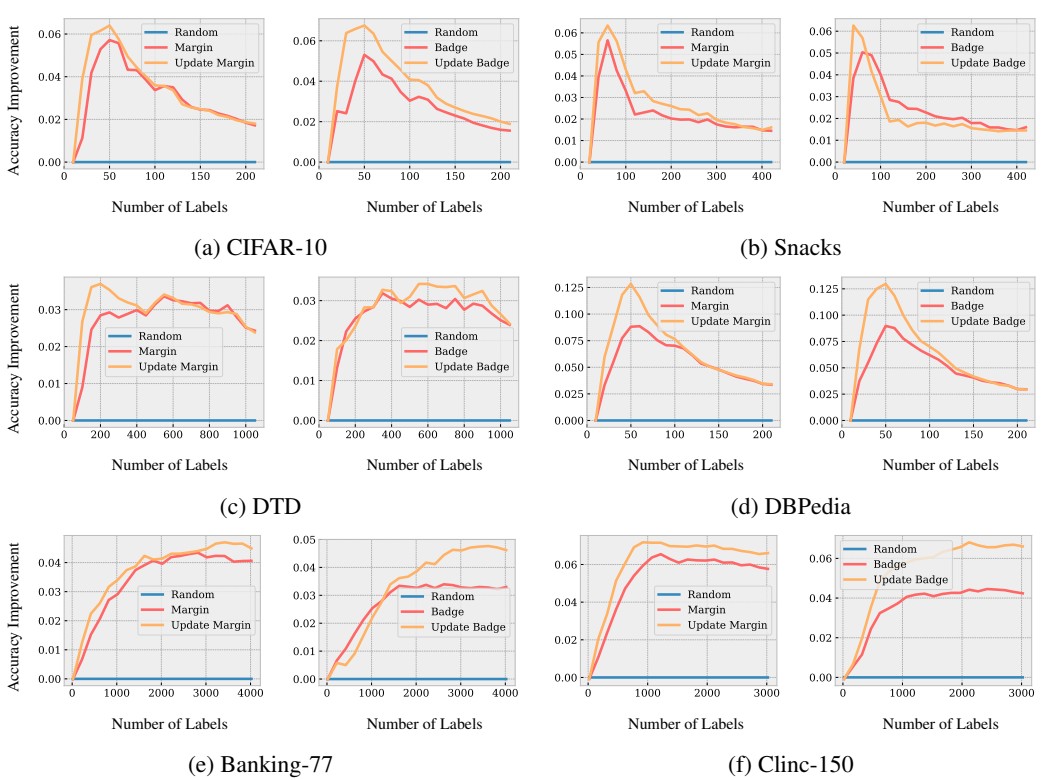

Figure 9: Accuracy improvement curves for different strategies and datasets showing the accuracy difference between the respective selection strategy and random instance selection.

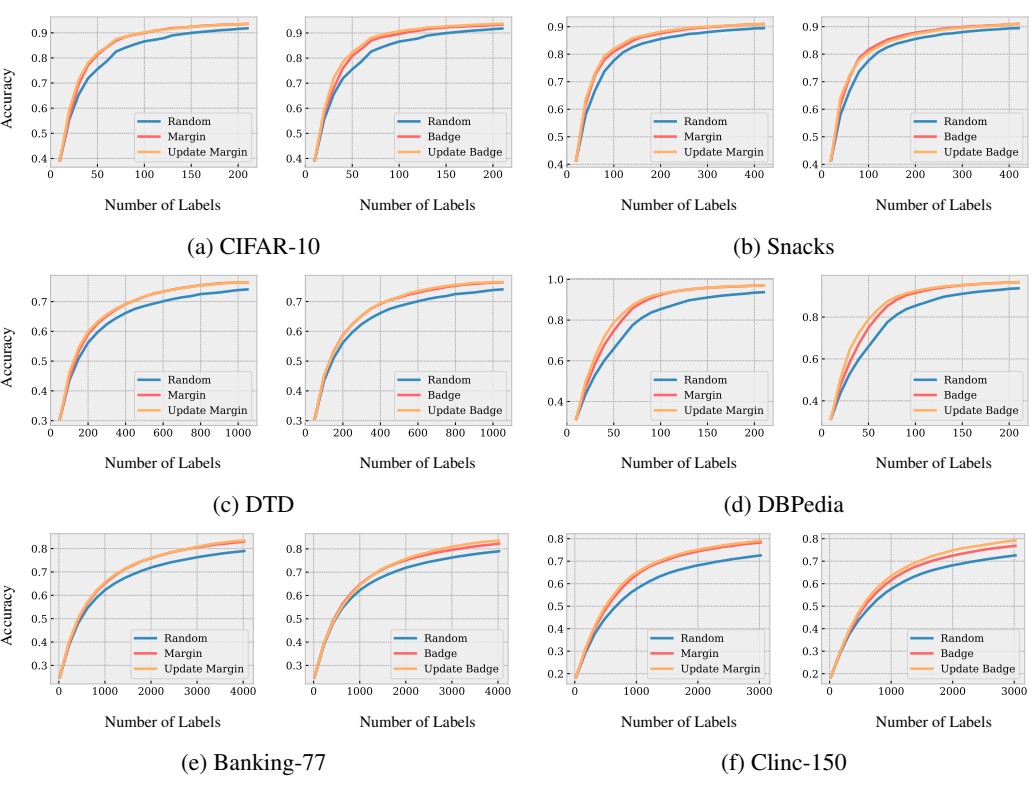

Figure 10: Learning curves for different strategies and datasets showing the accuracy.

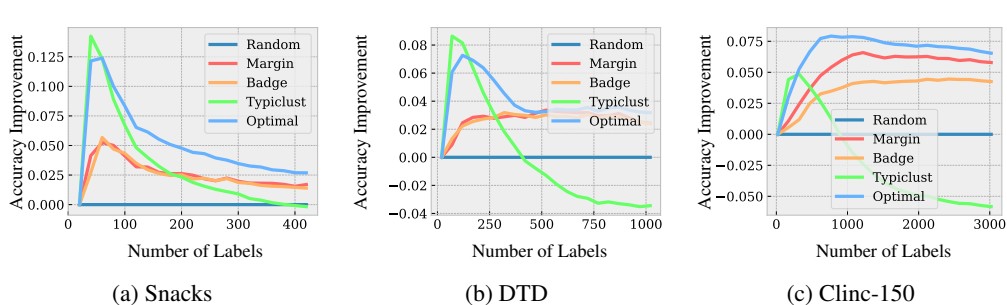

(a) Snacks      (b) DTD      (c) Clinc-150

Figure 11: Accuracy improvement over random selection of popular selection strategies compared to our upper baseline approximating optimal batch selection.

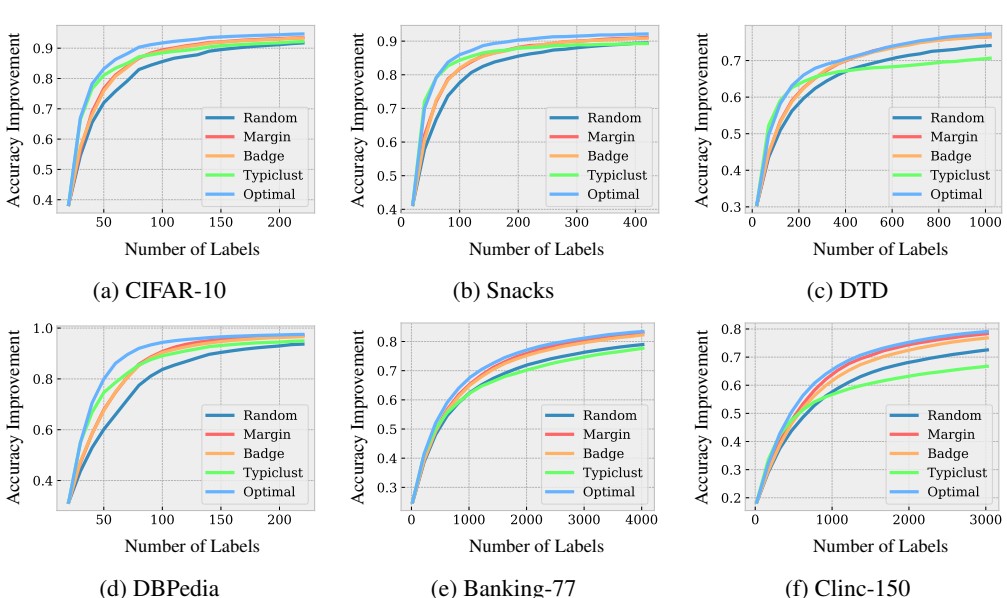

(a) CIFAR-10      (b) Snacks      (c) DTD

(d) DBPedia      (e) Banking-77      (f) Clinc-150

Figure 12: Learning curves reporting the accuracy of popular selection strategies compared to our upper baseline approximating optimal batch selection.

