# OpenReview forum: "Efficient Bayesian Updates for Deep Active Learning via Laplace Approximations"
_ICLR.cc/2025/Conference — ICLR 2025 Conference Withdrawn Submission_

### Official Review · Reviewer_mHji · 2024-10-28

**Soundness:** 3
**Presentation:** 3
**Contribution:** 3
**Rating:** 6
**Confidence:** 3

**Summary:**

This paper proposes a method to replace costly retraining with an efficient Bayesian update that uses a second-order optimization step based on a Gaussian posterior derived from a last-layer Laplace approximation. This approach achieves low computational complexity by computing the inverse Hessian in closed form, offering a solution that closely approximates full retraining but is significantly faster. Additionally, the new framework supports sequential batch selection by incrementally updating the DNN after each label acquisition. A look-ahead selection strategy is also introduced to provide an upper baseline for optimal batch selection.

**Strengths:**

Interesting perspective on improving computational efficiency on AL. This method does not need to recompute the Hessian from scratch. Instead, their updates use covariances through Laplace Approximations and the Woodbury identity for closed-form inversion. This is a very interesting work combining theoretical groundings and experimental evidence.

**Weaknesses:**

1. Lack of related work. For look ahead approaches, as the authors illustrate in Section 5.2, "The idea of look-ahead strategies is to select instances that, once labeled and added to the labeled pool, maximize the performance of the model", recently there are many data-driven approaches, which trains neural networks to predict the performance of labeled pool after adding certain instances [1] or uses well-calibrated probabilistic models to quantify the epistemic uncertainty about the unknown dynamics [5]. The authors shall do more broad search on related works.
2. For datasets choice, CIFAR-10 is a relatively easy datasets if the authors chose to use ViT as the classifier. For instance, [2] uses ViT on CIFAR100 and mini-ImageNet. I would expect more complicated datasets such as CIFAR100, mini-ImageNet, ImageNet scale of experiments. The authors should also report standard errors/deviations as many AL papers report it [1, 2, 3, 4].


References:

[1] Ding, Zixin, et al. "Learning to Rank for Active Learning via Multi-Task Bilevel Optimization." The 40th Conference on Uncertainty in Artificial Intelligence.

[2] Parvaneh, Amin, et al. "Active learning by feature mixing." Proceedings of the IEEE/CVF conference on computer vision and pattern recognition. 2022.

[3] Hacohen, Guy, et al. "Active Learning on a Budget: Opposite Strategies Suit High and Low Budgets." International Conference on Machine Learning. PMLR, 2022.

[4] Hübotter, Jonas, et al. "Transductive Active Learning with Application to Safe Bayesian Optimization." ICML 2024 Workshop: Aligning Reinforcement Learning Experimentalists and Theorists.

[5] Treven, Lenart, et al. "Optimistic active exploration of dynamical systems." Advances in Neural Information Processing Systems 36 (2023): 38122-38153.

**Questions:**

No.

---

### Official Review · Reviewer_WsVN · 2024-10-31

**Soundness:** 1
**Presentation:** 4
**Contribution:** 1
**Rating:** 5
**Confidence:** 4

**Summary:**

This paper considers an important problem in deep active learning by updating the model given the new queried data. Specifically, this paper examines an efficient Bayes update rule with experimental validations across various datasets.

**Strengths:**

This paper aims to tackle an important problem by efficiently updating the newly queried data in active learning. It’s well known that retraining from scratch is a bad idea since it unnecessarily consumes computational resources, while a naive update may lead to catastrophic forgetting problems. Based on these viewpoints, I believe this paper addresses an important issue in active learning. The paper is overall clearly written, with diverse experimental settings.

**Weaknesses:**

I have two major concerns regarding the paper. Therefore, I do not think the current paper is ready for publication.

- The first concern is a lack of discussion and comparison with a very related paper [1]. I believe these two issues in active learning have been clearly discussed, and different continual learning methods have been tested. When comparing paper [1] with the current submission, I would say paper [1] seems more comprehensive and useful in terms of practice, while the current submission only works within a Bayes-based active learning framework. In addition, paper [1] was accepted in 2023; therefore, I do not consider it a concurrent paper. Given the existence of paper [1], I think the novelty and significance of the current paper are rather incremental.

- My second issue is about the problem setting and formulation. In Lines 202 and 205, we assume that D^{+} is independent of D. In the first experiment (Sec 4), there is no problem with this. However, in the active learning setting, this assumption is fundamentally incorrect.
Let me explain the rationale:

1. The next query dataset D^{+} should definitely depend on the previous data D. Given an active learning algorithm A, the next query dataset should be a function of D^{+} = A(D, \text{current\_model}); therefore, there must be a dependency between the last and current datasets. This is also relevant to your discussion about online and continual learning, where both are not i.i.d. in this context.

2. To address this important issue, I would suggest the authors consider sequential Bayes (or related works in sequential RL). By clearly modeling the sequential data as D^{0}, D^{1} = \{D^{0}, D^{+, 1}\}, you can better reflect the actual interactive nature of active learning.


[1] Accelerating Batch Active Learning Using Continual Learning Techniques. TMLR 2023

**Questions:**

Please the weakness section and my suggestions.

---

### Official Review · Reviewer_SQEg · 2024-11-03

**Soundness:** 2
**Presentation:** 2
**Contribution:** 1
**Rating:** 3
**Confidence:** 5

**Summary:**

The authors propose a method for deep active learning based on a Bayesian approximation scheme.  The Bayesian approximation provides a computationally lightweight proxy instead of full model re-training.  This can enable a selection of a good batch of samples to label next within an active learning process.

**Strengths:**

The approach is based on a Gaussian approximation to the last layer of the network, and efficient approximations to update the distribution based on new labeled data.  The idea of using the final layer for data selection in active learning is central to other schemes such as Badge and LabelBench (see reference for the latter below). However, this particular Bayesian approach to deep active learning appears to be a bit different from past approaches.

The paper has two main components: 1) derivation and analysis of the Bayesian approximation; 2) experimental evaluation of the proposed approach. The experimental section of the paper contains two main parts: a) evaluation of the accuracy of the proposed Bayesian approximation and hyperparameter choices that affect the accuracy; b) evaluation of the performance of the proposed margin-based active learning scheme, which is based on the approximation.

**Weaknesses:**

Most of the paper is dedicated to the derivation the last layer Laplace approximation.  The approach is straightforward, and I do not feel this is a significant contribution to the field.  It is based on standard techniques and approximations.  There is nothing particularly novel or insightful about the approach.

The active learning approach is a standard margin-based procedure.  The novelty is that it uses the last-layer Laplace approximation rather than the actual model and margin.  This means that the last-layer Laplace approximation is the main contribution of the paper.  I wonder if there are other applications of the idea, beyond active learning.  The authors seem to consider a data selection application (“look-ahead” strategy) at the end of the paper.

The authors appear to be unaware of past work using lightweight proxies instead of full model retraining in deep active learning:

Coleman, C., Yeh, C., Mussmann, S., Mirzasoleiman, B., Bailis, P., Liang, P., ... & Zaharia, M. Selection via Proxy: Efficient Data Selection for Deep Learning. In International Conference on Learning Representations.

Zhang, Jifan, Yifang Chen, Gregory Canal, Stephen Mussmann, Yinglun Zhu, Simon Shaolei Du, Kevin Jamieson, and Robert D. Nowak. "LabelBench: A Comprehensive Framework for Benchmarking Label-Efficient Learning." arXiv preprint arXiv:2306.09910 (2023).

**Questions:**

The accuracy improvements shown in Figure 6 tend to decrease as the number of labels increases in some cases, but not others.  This a bit surprising and confusing, and I wonder if the authors could comment on this.

---

### Official Review · Reviewer_bqsP · 2024-11-03

**Soundness:** 2
**Presentation:** 3
**Contribution:** 1
**Rating:** 3
**Confidence:** 5

**Summary:**

The paper proposes a Bayesian approach to efficiently update neural network training during an active learning process. This is to reduce the heavy retraining cost of doing deep active learning. The authors propose a second order method through Laplace approximation on the last neural network linear layer. The method is evaluated both in terms of approximation accuracy and its usage in active learning.

**Strengths:**

The paper studies an important problem in active learning and the experiments demonstrates the method is effective compared to various active learning baseline algorithms. The use of Bayesian approximation is also new in the context of active learning. The experiments are conducted for both text and image datasets with various transformer architectures.

**Weaknesses:**

I have a couple major concerns for this paper:
1. There are many existing ways of efficiently approximating the model performance and reduce retraining cost in deep active learning. For example, Selection via proxy [1] shows using a smaller proxy model can be an effective alternative. Along that line, LabelBench [2] shows in one can simply retrain the last layer during selection to obtain the same performance in the end. In another line of research [3-5], the authors leverage NTK approximations to efficiently compute the look-ahead strategy. Yet another line of research, [6-7] uses warm start/continual learning approach to combat the retraining cost of neural networks in active learning. This paper does not compare or even mention any of these literature.

2. The Bayesian approximation approach has been applied in adjacent fields such as data selection [8]. The novelty of the approach is debatable.

[1] Coleman, Cody, Christopher Yeh, Stephen Mussmann, Baharan Mirzasoleiman, Peter Bailis, Percy Liang, Jure Leskovec, and Matei Zaharia. "Selection via proxy: Efficient data selection for deep learning." arXiv preprint arXiv:1906.11829 (2019).

[2] Zhang, J., Chen, Y., Canal, G., Das, A. M., Bhatt, G., Mussmann, S., ... & Nowak, R. D. (2024). LabelBench: A Comprehensive Framework for Benchmarking Adaptive Label-Efficient Learning. Journal of Data-centric Machine Learning Research.

[3] Mohamadi, M. A., Bae, W., & Sutherland, D. J. (2022). Making look-ahead active learning strategies feasible with neural tangent kernels. Advances in Neural Information Processing Systems, 35, 12542-12553.

[4] Wang, H., Huang, W., Wu, Z., Tong, H., Margenot, A. J., & He, J. (2022). Deep active learning by leveraging training dynamics. Advances in Neural Information Processing Systems, 35, 25171-25184.

[5] Wen, Z., Pizarro, O., & Williams, S. (2023). NTKCPL: Active Learning on Top of Self-Supervised Model by Estimating True Coverage. arXiv preprint arXiv:2306.04099.

[6] Ash, J., & Adams, R. P. (2020). On warm-starting neural network training. Advances in neural information processing systems, 33, 3884-3894.

[7] Das, A., Bhatt, G., Bhalerao, M., Gao, V., Yang, R., & Bilmes, J. (2023). Accelerating batch active learning using continual learning techniques. arXiv preprint arXiv:2305.06408.

[8] Deng, Z., Cui, P., & Zhu, J. (2023). Towards accelerated model training via bayesian data selection. Advances in Neural Information Processing Systems, 36, 8513-8527.

**Questions:**

See weakness.

---

### Official Review · Reviewer_q6ee · 2024-11-05

**Soundness:** 2
**Presentation:** 2
**Contribution:** 2
**Rating:** 3
**Confidence:** 3

**Summary:**

The paper introduces an efficient method for deep active learning by using a last-layer Laplace approximation to update the model without full retraining. The proposed method is tested on Bayesian updates where the new dataset is i.i.d. sampled, and then tested on experiments where the new data is obtained via active learning algorithms.

**Strengths:**

The proposed method is interesting as it can be potentially used for turning any sequential AL strategy into a batch version. Especially that the proposed method is efficient and computationally inexpensive.
Experimental results on different datasets demonstrate that the effectiveness of the proposed method.

**Weaknesses:**

There are some concerns about the soundness of the proposed method. Details follow.
* The proposed method establishes on the assumption that the original dataset $\\mathcal{D}$ and the newly acquired dataset $\\mathcal{D}^{\\oplus}$ are i.i.d., but this is false for actively acquired dataset.
* In section 4, it seems that there is a obvious baseline that is missing, i.e., fine-tuning the last layer on the newly-obtained dataset, the entire dataset, or a mixture of both. How would they compare to the proposed method?
* As shown in Figure 2, the proposed method is sensitive to the parameter $\\gamma$. I acknowledge that the paper fixes $\gamma=10$ across different datasets to be consistent. Does the optimal choice of the $\gamma$ sensitive to different scale of the model?

**Questions:**

* In section 5, since the proposed method needs to iteratively select the highest-scoring instance $b$ times, and Bayesian update the model for $b$ times. How does it compare to selecting $b$ instances in a single batch in terms of time cost?

---

### Note · Authors · 2024-11-13

**Comment:**

We have decided to withdraw our paper. We would like to thank everyone for their time and constructive feedback.

**Withdrawal Confirmation:**

I have read and agree with the venue's withdrawal policy on behalf of myself and my co-authors.